# Characterization and Expression Analysis of the *ALOG* Gene Family in Rice (*Oryza sativa* L.)

**DOI:** 10.3390/plants14081208

**Published:** 2025-04-14

**Authors:** Xi Luo, Hongfei Wang, Yidong Wei, Fangxi Wu, Yongsheng Zhu, Hongguang Xie, Huaan Xie, Jianfu Zhang

**Affiliations:** 1Rice Research Institute, Fujian Academy of Agricultural Sciences, Fuzhou 350019, China; luoxi@faas.cn (X.L.);; 2State Key Laboratory of Ecological Pest Control for Fujian and Taiwan’ Crops/Key Laboratory of Germplasm Innovation and Molecular Breeding of Hybrid Rice in South China/Fujian Engineering Laboratory of Crop Molecular Breeding/Fujian Key Laboratory of Rice Molecular Breeding/Fuzhou Branch, National Center of Rice Improvement of China/National Engineering Laboratory of Rice/South Base of National Key Laboratory of Hybrid Rice for China, Fuzhou 350003, China

**Keywords:** rice, ALOG, IDRs, RNA sequencing, inflorescence development

## Abstract

ALOG (*Arabidopsis* LSH1 and *Oryza* G1) proteins constitute a plant-specific family of transcription factors that play crucial roles in lateral organ development across land plants. Initially identified through forward genetic studies of Arabidopsis LSH1 and rice G1 proteins, ALOG family members have since been functionally characterized in various plant species. However, research focusing on the characteristics and expression patterns of all *ALOG* family members in rice remains relatively limited. In this study, we systematically characterized *OsALOG* family genes in rice. Compared to other genes in rice and *Arabidopsis*, the ALOG family genes have a relatively simple structure. The alignment of OsALOG amino acid sequences and analysis of disorder predictions reveal that all members possess conserved ALOG domains, while the conservation of intrinsically disordered regions (IDRs) is relatively low. Four amino acids—alanine, glycine, proline, and serine—are significantly enriched in the IDRs of each ALOG protein. Synteny analysis indicates that most *OsALOG* genes have undergone considerable divergence compared to their counterparts in *Arabidopsis*. Bioinformatic analysis of *cis*-regulatory elements predicts that *OsALOG* family genes contain elements responsive to ABA, light, and methyl jasmonate, although the abundance and composition of these elements vary among different members. The expression patterns associated with the rice floral development of *OsALOG* genes can be broadly categorized into two types; however, even within the same type, differences in expression levels, as well as the initiation time and duration of expression, were observed. These results provide a comprehensive understanding of the structural characteristics and expression patterns of *OsALOG* members in rice.

## 1. Introduction

In 2009, Yoshida et al. classified the Domains of Unknown Function 640 (DUF640) proteins in the Pfam protein family database as the ALOG (*Arabidopsis* LSH1 and *Oryza* G1) domain [1]. Research indicates that ALOG proteins exhibit characteristics such as sequence-specific DNA binding, transcriptional regulatory activity, and nuclear localization [2,3]. Specifically, ALOG proteins are a family of plant-specific transcription factors that play a crucial role in the development of lateral organs in land plants [4]. Increasing evidence suggests that ALOG proteins regulate inflorescence architecture and floral organ development through an inhibitory role. Currently, the functions of five *OsALOG* genes have been reported in rice. *OsG1*/*ELE* can repress the homeotic transformation of the sterile lemma to the lemma [5,6], while *OsG1* can affect the expression of various phytohormone-related genes by regulating critical transcription factors [7]. *OsG1L6/TH1/BLS1/BH1/AFD1* regulate the cell extension of the lemma and palea to determine grain shape and size, which may function as a transcription repressor and regulate downstream hormone signal transduction and starch/sucrose metabolism-related genes [8,9,10,11,12,13,14]. *OsG1L5/TAW1* is a unique regulator of meristem activity in rice and regulates inflorescence development through the promotion of inflorescence meristem activity and the suppression of the phase change to spikelet meristem identity [15]; *OsG1L1* and *OsG1L2* are likely crucial in regulating inflorescence architecture, serving as positive regulators for the development of primary branches and secondary branches. Their role in these processes is indicated by their expression in the reproductive meristems, following a pattern similar to *TAW1* [16].

In *Arabidopsis*, the functions of six *AtALOG* genes have been identified. *AtLSH1* was the first *ALOG* gene identified in plants, initially cloned from a dominant *Arabidopsis* mutant (lsh1-D). This mutant exhibits hypersensitivity to continuous far-red, blue, and red light, resulting in shorter hypocotyls compared to wild-type plants. *AtLSH1* plays a role in light regulation during seedling development, and its function relies on phytochromes [17]. Overexpression of *AtLSH1* and *AtLSH2* greatly inhibited hypocotyl elongation in a light-independent manner and reduced both vegetative and reproductive growth [18]. *AtLSH4* and *AtLSH3* are a pair of paralogous genes in *Arabidopsis* that can be directly activated and upregulated by the NAC family transcription factor *CUP-SHAPED COTYLEDON1* (*CUC1*). Meanwhile, *AtLSH4* and *AtLSH3* play a role in inhibiting organ differentiation at the boundary region, a crucial function during plant development [19]. Additionally, *AtLSH8* positively regulates ABA signaling by changing the expression pattern of ABA-responsive proteins [20]. The role of *AtLSH10* suggests that the OTLD1-LSH10 complex acts as a co-repressor, potentially representing a general mechanism for the specific function of plant histone deubiquitinases at their target chromatin [21].

*ALOG* gene functions have also been reported in other species. In tomato, the *ALOG* gene *TERMINATING FLOWER* (*TMF*) controls flowering by preventing the precocious expression of the floral identity gene *ANANTHA* [22]. In sorghum (*Sorghum bicolor*), the *ALOG* gene *DOMINANT AWN INHIBITOR* (*DAI*) inhibits awn elongation by suppressing both cell proliferation and elongation [23].

In this study, we systematically characterized the *OsALOG* family genes in rice using bioinformatic approaches, including phylogenetic analysis, protein structure analysis, synteny analysis, and *cis*-regulatory element analysis. A notable aspect of this study is the enrichment analysis of residues within intrinsically disordered regions (IDRs). Intrinsically disordered proteins (IDPs) constitute approximately 30% of eukaryotic proteomes [24]. Recent studies have increasingly revealed that IDPs are involved in essential functions such as transcription factor regulation, cell signaling, and molecular chaperone activity [25,26]. Moreover, IDPs play a pivotal role in driving protein phase transitions, a concept that has emerged as a key focus in biological research for understanding cellular compartmentalization and the regulation of biochemical reactions [27]. Additionally, we performed RNA-Sequencing (RNA-Seq) on nine sampling points from young panicles and spikelet organs in rice. We obtained expression profiles for 10 *OsALOG* genes from different inflorescence stages and spikelet organs. These studies offer a comprehensive understanding of the characteristics and expression patterns of *OsALOG* genes from a global perspective. Moreover, our results establish a foundation for further exploration of the molecular mechanisms underlying rice flower development by presenting RNA-Seq data from various inflorescence stages and organs.

## 2. Materials and Methods

### 2.1. Selection of Plant Materials and RNA Sample Collection

*Indica* rice cv. Huanghuazhan was planted in a greenhouse at the Rice Research Institute, Fujian Academy of Agricultural Sciences in Fuzhou, China. The seeds were sown on 20 May 2019, with transplanting conducted on 17 June 2019. The trial employed a randomized complete block design (RCBD) with three replicates. Each experimental plot comprised 10 rows of 7 hills per row, planted at a uniform spacing of 20 cm (within rows) × 20 cm (between rows) using single-seedling transplantation. The rice seedlings were raised via wet nursery cultivation, and field management followed conventional agronomic practices for paddy rice. We sampled young panicles at four stages and five spikelet organs at the booting stage from Huanghuazhan for tissue used for RNA-Seq and qRT-PCR. Detailed sampling points are shown in Table 1. After sampling, the tissues were snap-frozen in liquid nitrogen and stored at −70 °C until RNA extraction. Three biological replicates were used for each of the sampling points.

### 2.2. Identification and Phylogenetic Analysis of the ALOG Genes in Rice

For the purpose of identifying *ALOG* genes, we downloaded the genomic data of *Os.japonica*, *Os.indica*, and *Arabidopsis thaliana* (the assembly titles, respectively, are Os-Nipponbare-Reference-IRGSP-1.0 for the *Oryza sativa japonica* Group, Minghui63-Mhv2.0 assembly for the *Oryza sativa Indica* Group, and TAIR10 assembly for *Arabidopsis thaliana*) from the EnsemblPlants website (https://plants.ensembl.org/index.html) (accessed on 1 July 2024), and the hidden Markov model (HMM) files corresponding to the ALOG domain (PF04852) were downloaded from the Pfam protein family database (http://pfam.xfam.org/) (accessed on 1 July 2024) [29]. HMMER 3.4 [30] was used to search the *ALOG* genes from the genomic data of rice and *Arabidopsis*. Next, we extracted the protein sequences of the candidate *ALOG* genes by seqkit-2.8.2 software [31], and then determined their ALOG domains through the batch CD-search of NCBI (https://www.ncbi.nlm.nih.gov/cdd/?term=) (accessed on 2 July 2024). Phylogenetic trees were constructed using the Neighbor-Joining (NJ) method in MEGA11 [32] with the Poisson model, pairwise deletion, and 1000 bootstrap replications. Based on the results of domain prediction and phylogenetic analysis, redundant and incomplete sequences were manually removed.

### 2.3. Analysis of the Characteristics of the ALOG Family

Alignment of the ALOG protein sequences was performed using the ‘Muscle’ function in MEGA, with the Max Iteration set to 100. The aligned file was then refined and visualized using Jalview 2.11 [33]. The structures of the 30 identified *ALOG* genes were visualized using TBtools II v2.210 [34] based on the genome and gene annotation files. Prediction of the intrinsically disordered regions (IDRs) was carried out using DISOPRED3, a tool from the PSIPRED Workbench (http://bioinf.cs.ucl.ac.uk/psipred/) (accessed on 3 July 2024) [35]. To analyze the protein sequences of *OsALOG* genes, we used SnapGene 4.1.9 software to calculate the proportion of amino acids located within their intrinsically disordered regions (IDRs). To determine the enrichment of specific amino acids within the IDRs of each *OsALOG* gene, we performed an enrichment analysis using the ‘Simple Enrich’ function in TBtools II. After performing the analysis, we organized the resulting data in Excel, and then visualized the amino acid enrichment patterns using the ‘Heatmap’ function in TBtools II to compare the IDR composition across *OsALOG* genes. Motif discovery for ALOG proteins was performed using the MEME online tool (https://meme-suite.org) (accessed on 3 July 2024), with the number of motifs set to ten. Sequence length, molecular weights, isoelectric points, and subcellular location predictions for the identified ALOG proteins were obtained using tools from the ExPasy website (http://web.expasy.org/protparam/) (accessed on 4 July 2024) [36]. Prediction of transmembrane helices in ALOG proteins was conducted using TMHMM 2.0 on the DTU Health Tech website (https://services.healthtech.dtu.dk/services/TMHMM-2.0/) (accessed on 4 July 2024) [37].

### 2.4. Synteny Analysis

Gene duplication events were analyzed using the Multiple Collinearity Scan toolkit (MCScanX) with default settings [38]. The synteny relationships of the *ALOG* genes were visualized with TBtools II. Non-synonymous (Ka) and synonymous (Ks) substitutions for each duplicated *ALOG* gene were determined using KaKs_Calculator 2.0 [39].

### 2.5. Analysis of the Cis-Regulatory Elements

To analyze the promoter regions of *ALOG* genes, first use seqkit-2.8.2 to extract the 2000 bp sequence upstream of each gene’s ATG start codon. Next, submit these extracted promoter sequences to the PlantCARE database (http://bioinformatics.psb.ugent.be/webtools/plantcare/html/) (accessed on 5 July 2024) to identify *cis*-acting elements, and export the results in tabular format [40]. To visualize the distribution of these elements, load the annotated sequences into TBtools II and use the ‘Simple Biosequence Viewer’ feature. Finally, quantify and plot the *cis*-element counts by generating a heatmap using the ‘tidyverse 2.0’ package in R, with rows representing *ALOG* genes and columns representing element types.

### 2.6. RNA Extraction, Library Construction and Sequencing

Total RNA was isolated from each of the samples listed in Table 1 (including three biological replicates per sample) using the TRIzol Kit (Promega, Madison, WI, USA) following the manufacturer’s instructions. Then, the total RNA was treated with RNase-free DNase I (Takara Bio, Kusatsu, Shiga, Japan) for 30 min at 37 °C to remove residual DNA. RNA quality was verified using a 2100 Bioanalyzer (Agilent Technologies, Santa Clara, CA, USA) and was also checked by RNA electrophoresis on RNase-free agarose gels. The library construction and sequencing were performed according to previously established methods [7].

### 2.7. Quantitative Reverse Transcription-PCR Analysis

Total RNA was reverse transcribed using the ReverTra Ace qPCR RT Master Mix with the gDNA Remover kit (Toyobo, Japan). The resulting cDNA was diluted 1:20 and served as the template for qPCR. PCR reactions were set up with FastStart Universal SYBR Green Master (ROX) (Roche, Indianapolis, IN, USA) and conducted on a LightCycler 480II instrument (Roche, https://www.roche.com.cn/). The data were quantified and normalized against the endogenous control gene *UBQ5* using the 2^−ΔΔCT^ method [41]. The primer sequences for *OsALOG* cDNAs are listed in Supplemental Appendix A.

## 3. Results

### 3.1. Genome-Wide Identification and Phylogenetic Analysis of the ALOG Genes in Rice

Using the profile hidden Markov model of the *ALOG* gene family, we scanned the genomes of rice (*Oryza sativa japonica* and *Oryza sativa indica*) and *Arabidopsis*, identifying 10 genes containing the ALOG domain in each genome. We extracted the gene and amino acid sequences of the 30 *ALOG* genes for further analysis (Appendix A).

We examined the evolutionary relationships among the ALOG family members in rice and *Arabidopsis* by constructing an unrooted phylogenetic tree using MEGA. The 30 ALOG members were clearly divided into three groups, with Group I further subdivided into two subgroups (Figure 1A). Group I includes seven *OsALOG* genes, *OsG1*, *OsG1L1*, *OsG1L2, OsG1L3, OsG1L4, OsG1L5*, and *OsG1L6*, and four *AtALOG* genes, *AtLSH1*, *AtLSH2*, *AtLSH3*, and *AtLSH4*. Previous studies have shown that *OsG1* functions to suppress the overgrowth of sterile lemmas [1,5,6], while *OsG1L1*, *OsG1L2,* and *OsG1L5* are involved in regulating the number of secondary branches [15,16], *OsG1L6* is responsible for the development of lemmas and paleas in rice [9,10,11,12,13,14]. Additionally, overexpression of *AtLSH1* and *AtLSH2* greatly inhibited hypocotyl elongation in *Arabidopsis* [18]. *AtLSH4* and *AtLSH3* suppress organ differentiation in boundary regions, a critical regulatory mechanism for proper plant development [19]. The findings from these studies suggest that the *ALOG* genes in Group I share similar functions, particularly in suppressing the development of lateral organs. At present, the molecular functions of Group II ALOG genes remain unexplored. Additionally, Group III is specific to Arabidopsis. Taken together, the clustering results indicate that some *ALOG* family genes have retained ancestral features common to both monocots and dicots, while others have diverged to form species-specific gene structures.

### 3.2. Gene Structures, Protein Domains, and Motif Compositions of the Rice ALOG Gene Family

We visualized the gene structures, protein domains, and motif compositions of ALOG family members according to the clustering group order (for detailed information, please refer to Appendix A). Figure 1A shows that, with the exception of *OsG1L1* and *AtLSH4*, other *ALOG* genes possess a single coding sequence exon. Seventeen *ALOG* genes lack introns, eight genes contain one intron, and five genes (all from rice) harbor two introns. Overall, *ALOG* genes display structural simplicity (fewer exons) compared to the 4–6 exon average in rice and 4–5 average in Arabidopsis genes. (Figure 1A). Next, we performed domain predictions on 30 ALOG proteins. The results indicated that the ALOG domains of these 30 proteins are relatively conserved, with intrinsically disordered regions (IDRs) distributed at both ends of the amino acid sequences (Figure 1B). Among the ten OsALOG proteins (*Os. japonica*), OsG1L1 has the shortest protein length (191 aa), while OsG1L9 has the longest protein length (284 aa) (Appendix A). Amino acid sequence alignment of 30 ALOG family proteins also show a highly conserved ALOG and nuclear localization signal domains (Appendix A). However, we noticed that the protein of *OsG1* or *OsMH63G1* has two additional amino acid sequence regions in the ALOG domain compared to other ALOG family proteins in either rice or *Arabidopsis* (Appendix A). The variations in protein structure may lead to a distinct mechanism by which OsG1 regulates downstream genes to differ from other ALOG members.

In addition, we observed the insertion of a 237bp sequence in the coding region of *OsMH63G1L1*, which causes its protein sequence to be 79 amino acids longer than that of its homolog OsG1L1(Figure 1A, Appendix A), while other *Os.indica* varieties, such as 9311, do not show this difference in their *OsG1L1* homologous genes (Appendix A). This finding indicates that *OsMH63G1L1* has undergone an insertion mutation, but it does not affect its ALOG domain (Figure 1B). Further research is needed to determine whether this sequence variant affects the gene’s function.

To gain a detailed understanding of ALOG protein structures, we performed MEME motif analysis (for detailed motif information, please refer to Appendix A). Our analysis revealed that all 30 ALOG proteins contain conserved regions known as motif1, motif2, and motif3 within the ALOG domain. When comparing two rice subspecies (*Os.japonica* and *Os.indica*), we found that, with the exception of OsG1L1 and OsMH63G1L1— which differ by three motifs— the remaining nine pairs of ALOG paralogous proteins in rice show no differences. The other seven motifs are located within the IDRs of the ALOG proteins in rice or *Arabidopsis* (Figure 1C). This distribution suggests that the various combinations of these motifs within the IDRs categorize different ALOG proteins, allowing them to perform distinct biological functions possibly.

We also analyzed the physicochemical properties of the ALOG proteins. Compared to *Arabidopsis*, the ALOG proteins in rice exhibit a higher average number of amino acids and a greater average molecular weight. The isoelectric point of OsALOG proteins is 9.35, which is slightly lower than the isoelectric point of AtALOG proteins, which is 9.81 (Appendix A). Our predictions of subcellular localization indicate that all OsALOG proteins are localized in the nucleus. Additionally, we assessed the presence of transmembrane helices in ALOG proteins, and our results showed that these proteins do not contain any transmembrane helices (Appendix A).

### 3.3. Enrichment Analysis of Residues in IDRs

Previous studies have found that tomato ALOG paralogous proteins can form transcriptional condensates through IDR-driven phase separation, thereby regulating the expression of floral identity genes [22]. Moreover, the composition of IDRs and phase separation capacity correspond to the transcriptional repression ability of ALOG proteins [22]. Therefore, we investigated whether the amino acid residues in the IDRs of rice ALOG proteins exhibit any bias patterns. To this end, we performed a statistical analysis of the proportions of each amino acid residue across ten OsALOG-IDRs of rice (Figure 2). The proportion of non-polar amino acid residues in the OsALOG-IDPs exceeded 50%. Specifically, OsG1L4 had the lowest proportion at 50.67%, while OsG1L7 had the highest at 65.11%. Four amino acid residues were notably abundant: three non-polar amino acids—alanine, glycine, and proline—and one polar amino acid—serine. These four amino acids were also significantly enriched relative to the total length of the OsALOG proteins (Appendix A). We also assessed the net charge of ALOG-IDRs at pH 7.5. All OsALOG-IDPs exhibited basic charges except for OsG1L3 and OsG1L9, which displayed acidic charges.

### 3.4. Synteny Analysis of OsALOG Genes

We analyzed the syntenic relationship of *OsALOG* genes. As shown in Figure 3A, there are five pairs of segmental duplication events, unevenly distributed across chromosomes 01, 02, 04, 05, 06, 07, and 10. These results imply that segmental duplication events could play a crucial role in the expansion of the *ALOG* gene family. However, no tandem duplication events were identified. Among the 10 *OsALOG* genes, no segmental duplication events were detected for *OsG1*, *OsG1L6/TH1*, or *OsG1L9*, indicating that these genes uniquely evolved in rice.

To further understand the evolutionary characteristics of the *ALOG* family, we constructed comparative syntenic maps of rice and *Arabidopsis*. As shown in Figure 3B, four orthologous gene pairs have been found between rice and *Arabidopsis*.

Next, we calculated the Ka/Ks ratios of the *ALOG* syntenic gene pairs. The results show that Ka/Ks < 1 of the orthologous gene pairs *OsG1L7* and *AtLSH6*. This suggests that the function of *OsG1L7* has undergone a more rigorous purifying selection during evolution. However, the other 3 *ALOG* syntenic gene pairs showed a high sequence divergence value (Appendix A). This result suggests that, although these gene pairs are syntenic, they may have undergone significant structural and functional changes during the evolutionary divergence between rice and *Arabidopsis*.

### 3.5. Cis-Regulatory Elements of OsALOG Genes Analysis

Bioinformatic analysis predicted that the promoter regions of *OsALOG* genes contain multiple potential *cis*-regulatory elements putatively associated with stress responses and hormone signaling (Figure 4A). We focused on identifying the *cis*-elements enriched in the promoter regions of rice *ALOG* genes. Most *ALOG* promoter regions show a significant presence of ABRE elements related to ABA responsiveness, as well as G-box elements linked to light responsiveness. Furthermore, they contain numerous MeJA-responsive elements, including the CGTCA motif and the TGACG motif. The composition of *cis*-elements in *OsG1* differs from other ALOG members; it encompasses three CAT-box elements associated with meristem expression and three GT1-motif elements associated with light responsiveness. However, it contains only one ABRE element and two G-box elements. Additionally, the promoter region of *OsG1L6 (TH1)* harbors an RY-element related to seed-specific regulation, a characteristic not observed in other *OsALOG* genes (Figure 4A). Overall, the composition of *cis*-elements in *OsALOG* genes exhibits both commonalities and specificities, leading to distinct expression patterns depending on different internal or external environments.

### 3.6. Expression Pattern of OsALOG Genes

To investigate the expression characteristics of *OsALOG* genes during different stages of rice panicle development, we performed RNA-Seq analysis on young panicles (four stages) and spikelet organs (five tissues) at the booting stage from the *indica* rice cultivar Huanghuazhan (Table 1). In total, 27 cDNA libraries were constructed and sequenced. The RNA-Seq data were uploaded to the Sequence Read Archive (SRA) of the National Center for Biotechnology Information (accession number PRJNA1148009).

The *OsALOG* expression profiles were generated using transcriptome data across nine sampling points, with quantitative reverse transcription PCR (qRT-PCR) validation confirming the reliability of the RNA-seq results (Figure 5 and Figure 6). The results indicate that the 10 *OsALOG* genes are distinctly divided into two major groups: group Ⅰ and group Ⅱ. Group Ⅰ includes *OsG1*, *OsG1L1*, *OsG1L2*, *OsG1L5*, and *OsG1L6*. These genes exhibit high expression levels during the stage of young panicle differentiation but low or no expression in the spikelet organs at the booting stage. Group Ⅱ includes *OsG1L3*, *OsG1L4*, *OsG1L7*, *OsG1L8*, and *OsG1L9*. Unlike group I, these genes are not specifically expressed during the young panicle stage, but are dispersedly expressed across each sampling point (Figure 5).

The results of qRT-PCR demonstrated that *OsG1*, *OsG1L2*, and *OsG1L5* share a similar expression pattern, with their expression levels gradually decreasing from the S1 to S4 stages of young panicle differentiation. In contrast to the expression patterns of the three genes mentioned above, *OsG1L6* exhibits a steady rise in expression from the S1 to S4 stages, while *OsG1L1* shows an initial increase followed by a decrease (Figure 6). Based on FPKM (Fragments Per Kilobase Million) values (Appendix A), the overall expression levels of the genes from group Ⅰ are ranked as *OsG1* > *OsG1L2* > *OsG1L6* > *OsG1L1* > *OsG1L5*.

Although most genes from group II exhibit relatively high expression levels in spikelet organs, each gene has a specific expression pattern. Specifically, *OsG1L4* exhibits relatively high expression levels in the palea and inner floral organs; *OsG1L3* shows relatively high expression levels in the S1 stage and the lemma; *OsG1L9* has relatively high expression levels at the S1 stage or in the sterile lemma; *OsG1L7* and *OsG1L8* are expressed at all nine sampling points, with overall expression significantly higher in spikelet organs compared to the young panicle stage (Figure 6, Appendix A). The expression patterns and biological functions of *ALOG* genes within Group II still require further investigation.

## 4. Discussion

Regarding the function of ALOG family proteins, they are generally considered to be transcription factors that regulate the development of floral organs in plants [2,3,4]. In this study, we compared the amino acid sequences of ALOG proteins and found that the ALOG domain is quite conserved, while the amino acid sequences in the IDRs show poor conservation across ALOG orthologs (Appendix A). Studies have shown that the sequence-ensemble relationships of IDPs directly influence their biological functions, and these relationships are governed by the amino acid composition and sequence patterns within the IDPs [42,43]. Therefore, we infer that the functions of ALOG family proteins have diverged and are influenced by IDPs. In fact, previous research suggests that the conserved serine bias in the IDRs of the yeast transcription factor MED1 is essential for its phase separation capacity [44]. Additionally, it has been suggested that the enrichment of polar amino acids indicates a protein’s potential for phase separation [45,46,47]. However, in this study, we found that non-polar amino acids are significantly enriched in OsALOG-IDRs, while polar amino acids are only significantly enriched in serine. These findings imply that the residue composition preferences of IDPs could differ across various proteins. Similarly, the residue composition preferences of ALOG-IDRs may differ across species. Therefore, further in-depth research on this topic will be needed in the future.

Currently, the functions and expressions of five *OsALOG* family genes have been studied: *OsG1*, *OsG1L1*, *OsG1L2*, *OsG1L5*, and *OsG1L6*. In this study, the transcript expression of ten rice *ALOG* family genes was analyzed. Coincidentally, the expression profiles of the five genes are classified into one group (group I). Generally, these genes are highly expressed during the differentiation stage of young panicles and are either weakly expressed or not expressed at all in the organs of spikelets during the booting stage. This expression pattern is largely consistent with previous studies. Existing research has shown that transcripts of *OsG1* are preferentially accumulated in the sterile lemma during the stages of first and secondary rachis branch primordia emergence, continuing through to the stamen primordia formation stage. The expression decreases as the spikelets develop and eventually disappear in the sterile lemma of nearly mature flowers [6], and *OsG1L5/TAW1* mRNA continued to accumulate in inflorescence meristem until the initiation of the primary branch primordia. *TAW1* expression gradually disappeared [15], while *OsG1L1* and *OsG1L2* have a similar expression pattern to *OsG1L5/TAW1* [16]. However, based on the RNA-seq and qPCR data from this study, there are still some differences in the expression forms of *OsG1L1*, *OsG1L2*, and *OsG1L5/TAW1*. To be specific, *OsG1L2* shows significantly higher transcript levels than the other two genes during young panicle differentiation. Moreover, compared to other *OsALOG* genes, *OsG1L2* maintains considerable expression abundance in spikelet organs at the booting stage. *OsG1L1* shows low expression in the sterile lemma, palea, and lemma, with almost no expression in other spikelet organs. On the other hand, *OsG1L5/TAW1* is expressed only during the young panicle stage and is almost not expressed in the spikelet organs (Appendix A). *OsG1L6* is also expressed during the young panicle differentiation stage. Unlike the other genes in Group I, *OsG1L6* expression gradually increases across the four sample points. Additionally, *OsG1L6* is expressed at lower levels in the palea and lemma during the booting stage, with no expression detected in other floral organs.

Based on this study and previous research, we have speculated the expression patterns of the five genes during the young panicle development stage as follows: *OsG1L5* begins to express when the inflorescence primordium appears and ceases expression by the time the primary branch primordium emerges. Both *OsG1L1* and *OsG1L2* also start expressing when the inflorescence primordium appears, but their expression persists for a longer duration. *OsG1L1* continues to show some expression in the spikelet meristem, while *OsG1L2* even maintains some expression in the spikelet organs during the booting stage. *OsG1* begins to express after the expression of the *OsG1L5*, *OsG1L1,* and *OsG1L2*, in other words, it starts when the primary branch primordium appears, and stops expressing after spikelet differentiation is complete. Currently, there is no evidence of *OsG1L6* expression in the inflorescence meristem. Furthermore, existing studies have shown that *OsG1L6* mutants exhibit no observable phenotypic alterations in branch formation or sterile lemma and rudimentary glume development [8,9,10,11,12,13,14]. Based on these observations, we suggest the possibility that *OsG1L6* transcript accumulation follows *OsG1* expression, terminating when spikelet differentiation is complete. Compared to group I, research on the biological functions of *OsALOG* genes in group II is limited. In this study, the expression profiles of *OsALOG* genes in group II display distinct characteristics. *OsG1L3* and *OsG1L4* exhibit significantly different expression patterns compared to *OsG1L5*. However, synteny analysis within the rice genome shows that *OsG1L3*, *OsG1L4*, and *OsG1L5* are collinear gene pairs with each other. We suspect these differences result from upstream transcriptional regulation. Promoter region analysis reveals substantial differences in *cis*-elements associated with phytohormones between *OsG1L5* and the other two genes, suggesting that their expression is regulated by transcription factors related to phytohormones. *OsG1L7* and *OsG1L8* show similar expression patterns and were identified as collinear pairs through synteny analysis. Additionally, the Ka/Ks ratio between *OsG1L7* and *AtLSH6* (*Arabidopsis* ortholog of *OsG1L7*) is less than 1, suggesting that *OsG1L7* originated before the divergence of monocots and dicots, and plays a crucial role in the evolutionary history of *ALOG* genes.

In conclusion, our analysis reveals that the *OsALOG* gene family exhibits both conserved and divergent characteristics in rice. All members of the OsALOG gene family possess the conserved ALOG domain, demonstrating their evolutionary conservation, while the intrinsically disordered regions (IDRs) at both the N-terminal and C-terminal ends reflect the divergence among individual ALOG members. Detailed explanations are provided below. Sequence alignments demonstrate that all members contain conserved ALOG domains, which confer their fundamental function as transcriptional repressors regulating lateral organ development. However, divergence in promoter architectures and intrinsic disorder regions (IDRs) generates distinct expression patterns and functional specialization among family members. We propose that *OsALOG* genes operate through a coordinated regulatory network, where individual members exhibit either complementary or antagonistic expression profiles (e.g., some genes are highly expressed in early floral meristems while others are specifically activated in organ primordia). This spatiotemporal regulation precisely orchestrates the development of spikelets, glumes, and inner floral organs, ultimately contributing to morphological diversity both within and between rice species. For future investigations, we highlight two research priorities: (i) the functional implications of amino acid residue preferences in ALOG-IDRs, and (ii) the precise biological functions of group II *OsALOG* genes(classified in expression profiles) during flower development.

## Figures and Tables

**Figure 1 plants-14-01208-f001:**
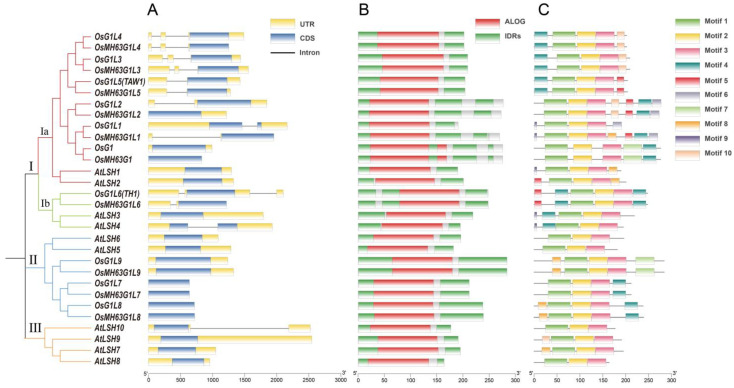
Characterization analysis of 30 ALOG family members in rice and *Arabidopsis*. (**A**) The phylogenetic tree of ALOG members and structures of ALOG genes. (**B**) Protein domains of ALOG members. (**C**) MEME motif analysis of ALOG proteins. Details of clusters are shown in different colors. Details of the motifs can be found in Appendix A. The length of genes and proteins can be estimated using the scale at the bottom.

**Figure 2 plants-14-01208-f002:**
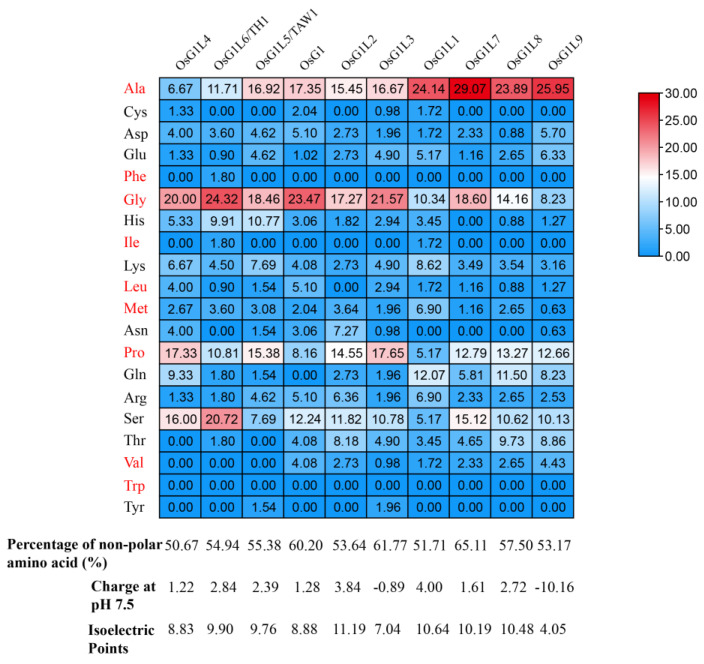
The proportions of amino acid residue in OsALOG-IDRs. The red text represents non-polar residues.

**Figure 3 plants-14-01208-f003:**
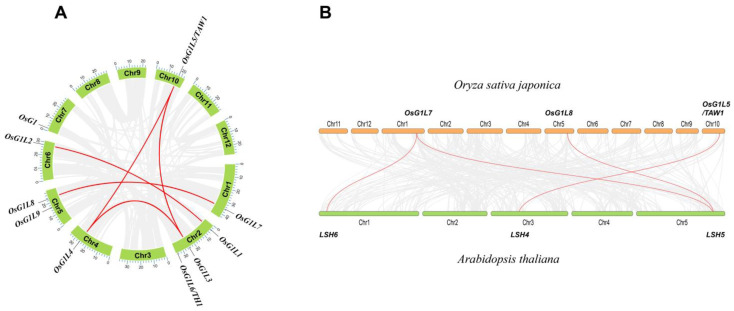
Synteny analysis of *OsALOG* genes. (**A**) Synteny analysis and distribution of *OsALOG* genes in rice (*Oryza sativa japonica*). (**B**) Comparative syntenic maps of *ALOG* genes in rice (*Oryza sativa japonica*) and *Arabidopsis*. Red line indicates homologous gene pairs.

**Figure 4 plants-14-01208-f004:**
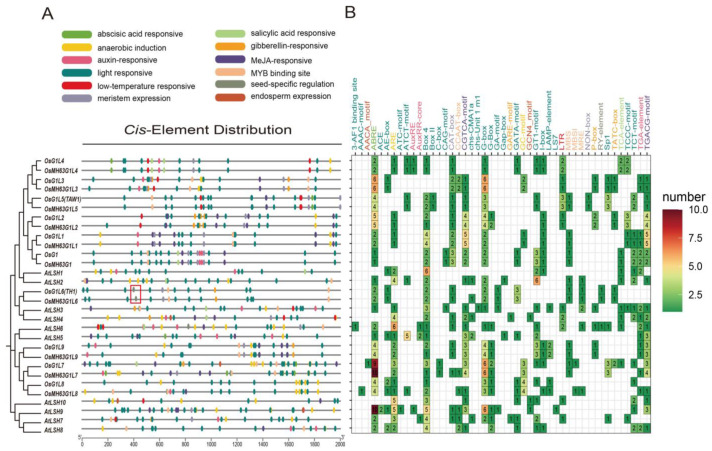
*cis*-regulatory elements of *ALOG* gene analysis. (**A**) Distribution of *cis*-regulatory elements in the promoter regions. (**B**) Heatmap showing the counts of *cis*-regulatory elements in the promoter regions. In Figure (**A**), the red box highlights the RY-element, while in Figure (**B**), different colored fonts of the elements indicate their corresponding annotations with Figure (**A**).

**Figure 5 plants-14-01208-f005:**
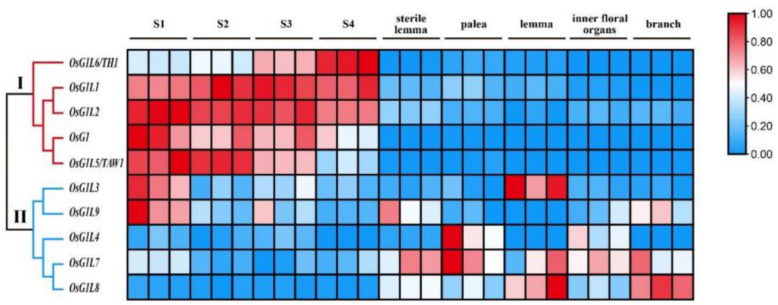
Expression profiles of the *OsALOG* genes in young panicles at four developmental stages and in five spikelet organs at the booting stage. The three data blocks below the black solid line represent three biological replicates. Details of the clusters are shown in different colors. FPKM values were normalized (Z-score) according to the row scale.

**Figure 6 plants-14-01208-f006:**
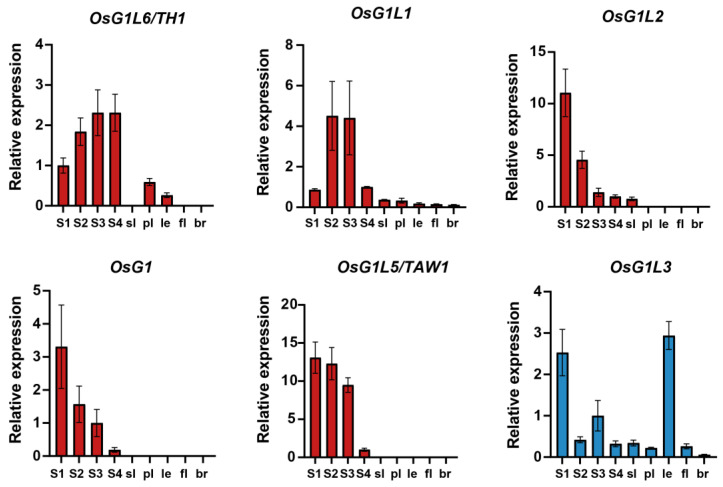
Expression analysis of 10 *ALOG* genes of nine sample points (three biological replicates) by qRT-PCR. Data were normalized to the *UBQ5* gene, and vertical bars indicate standard deviation. The fill colors of the bar (red and blue) correspond to group Ⅰ and group Ⅱ in Figure 5, respectively. S1–S4, four stages of young panicle; sl, sterile lemma; pl, palea; le, lemma; fl, inner floral organs; br, branch.

**Table 1 plants-14-01208-t001:** Description of samples collected for RNA-Seq at different panicle development stages.

Sample ID	Sampling Point	Length of Inflorescence	Apical Spikelet Development Stage
S1	Young panicle	1~2 mm	Formation of palea, elongation of Sterile lemma and lemma.
S2	Young panicle	2~5 mm	Formation of stamen and pistil primordia.
S3	Young panicle	5~10 mm	Stamen and pistil begin to differentiate, Palea and lemma close up gradually.
S4	Young panicle	10~15 mm	Formation of pollen and ovule.
sl	Sterile lemma	-	Booting stage
pl	Palea	-	Booting stage
le	Lemma	-	Booting stage
fl	Inner floral organs	-	Booting stage
br	Branch	-	Booting stage

Note: the determination of the apical spikelet development stage reference to previous research [28].

## Data Availability

The RNA-Seq datasets used in the current study are available in the PRJNA1148009 to SRA (Sequence Read Archive) repository (https://www.ncbi.nlm.nih.gov/bioproject/PRJNA1148009) (accessed on 5 July 2024).

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
