# Peer review of "Characterization and Expression Analysis of the ALOG Gene Family in Rice (Oryza sativa L.)"

_plants, 2025, doi:10.3390/plants14081208_

Round 1
Reviewer 1 Report
Comments and Suggestions for Authors
The manuscript by Luo et al. presents a detailed characterization and expression analysis of the ALOG family genes in rice (Oryza sativa L.). The study includes phylogenetic analysis, protein structure analysis, synteny analysis, cis-regulatory element analysis, and RNA-Seq-based expression profiling of OsALOG genes. The authors also analyzed the intrinsically disordered regions (IDRs) of ALOG proteins and the cis-regulatory elements in the promoter regions of OsALOG genes.
Overall, the manuscript is well-written, and the study is well-designed. The results provide a comprehensive understanding of the structural characteristics and expression patterns of OsALOG members in rice. The manuscript also provides a valuable resource for further research on the molecular mechanisms underlying rice flower development.
However, I have a few suggestions for improvement:
- The authors could provide a more detailed description of the biological significance of their findings. For example, how might the differences in expression patterns of OsALOG genes contribute to the diversity of rice flower development?
- The authors could discuss the limitations of their study and suggest future research directions. For example, the study is based on a single rice cultivar. It would be interesting to investigate the ALOG gene family in other rice cultivars.
- In the phylogenetic analysis, the authors could consider using other methods in addition to the Neighbor-Joining method.
- The authors could provide more information about the functions of the genes in Group II ALOG genes.
Author Response
- The authors could provide a more detailed description of the biological significance of their findings. For example, how might the differences in expression patterns of OsALOG genes contribute to the diversity of rice flower development?
Response: Thank you for your positive feedback. We appreciate your first suggestion regarding the need for a more detailed explanation of the biological significance of OsALOG genes. We propose that different OsALOG members, through complementary or antagonistic expression patterns (e.g., some genes are highly expressed in early floral meristems while others are specifically activated in organ primordia), form a hierarchical regulatory network that precisely controls the development of rice spikelets, glumes and inner floral organs, ultimately contributing to intraspecific or interspecific differences in floral morphology.
This explanatory text has been incorporated into the concluding paragraph of the discussion section, with appropriate highlighting (Page 12, Lines 433-438).
- The authors could discuss the limitations of their study and suggest future research directions. For example, the study is based on a single rice cultivar. It would be interesting to investigate the ALOG gene family in other rice cultivars.
Response: Thank you for your suggestion and the manuscript now states future research priorities (Page 12, Lines 439-442), with changes clearly highlighted in the revised version.
- In the phylogenetic analysis, the authors could consider using other methods in addition to the Neighbor-Joining method.
Response: We appreciate your suggestion. Using the Maximum Likelihood method, we reconstructed the phylogenetic tree and obtained results consistent with our previous findings.
- The authors could provide more information about the functions of the genes in Group II ALOG genes.
Response:Thank you for your suggestion. This study primarily focuses on the gene structure, protein characteristics, and expression profiles of the ALOG gene family in rice. We acknowledge that the functional aspects of Group II ALOG genes remain largely unexplored, which in fact represents a key direction for our future research.

Reviewer 2 Report
Comments and Suggestions for Authors
Plants Manuscript #: 3542680
Authors: Xi Luo et al., 2025
Title: Characterization and Expression Analysis of ALOG Family Members in Rice (Oryza sativa L.)
The authors have bioinformatic of gene and predicted proteins along with RNA expression (RNA-seq and qRT-PCR) data for the family of ALOG (Arabidopsis LSH1 and Oryza G1) genes and proteins in rice. They cite previous reports from a variety of groups that suggest the function of these genes/proteins are transcriptional regulators of development in rice. The authors themselves do not provide additional “functional” data, but instead provide bioinformatic phylogenetic, gene structure, predicted promoter regulatory cis-elements, general protein structure, and amino acid bias data that might relate to function and functional differences across the 10 genes in rice. Further, they provided RNA expression data (both RNA-seq and confirmatory qRT-PCR data) for the developmental expression (at the RNA level) for the same 10 genes in Indica rice (did not test Japonica rice) and in conclusion related expression patterns to phylogenetic, protein structure/aa sequence groups and compared to predicted function from cited reports.
Understanding rice development is critical for basic plant biology reasons and for the world-wide importance of rice for human nutrition, thus there is value in understanding these genes and the encoded proteins. In general, the methods and data analysis of the authors were reasonable and consistent with standard practices, but there were no profoundly new discoveries or conclusions as to function and roles of ALOG genes/proteins. However, the provided gene/protein characterizations and expression data do add some to the understanding of ALOG’s in rice. I take issue with some minor data interpretation of the expression data by the authors in the Discussion compared to what the provided data show, as mentioned below. On a positive note, it was good to see that the RNA-seq data closely matched the qRT-PCR data and that the authors did the important confirmation with the qRT-PCR analysis.
The biggest general scientific concern I have is that there are functional data (from previously cited reports) only for the Phylogenetic Group 1 genes/proteins (see Figure 1) that show them to be transcriptional factors, and no functional data (here or in other reports) for Phylogenetic Group 2 (rice and Arabidopsis genes) or Group 3 (Arabidopsis genes only). The authors seem to be making the general assumption that Group 2 and 3 proteins would also be transcription factors, but again no data to show that. Any additional functional data/evidence information about Phylogenetic Groups 2 and 3 genes/proteins would help this paper significantly. A second general issue that comes up throughout manuscript text, figures, and supplemental data is that authors should make it clear what Os genes / proteins are from Indica and which are from Japonica. This would help reader know if genes being discussed are orthologs or paralogs.
In addition to this concern, there a number of writing issues/errors, missing methods information, figure layout problems, and some gene expression data interpretation that I feel the authors need to resolve before this could be published. See my specific comments below for these issues.
Abstract:
Lines 15 - 17: It would help readers if some the predicted function(s) for transcription factors for ALOG proteins is mentioned in Abstract.
Line 20: Here “ALOG” is refereeing to genes, so it should be italicized.
Lines 26: Mention that these are predicted “promoter” elements in the genes, versus elements that might control RNA process or some other genetic process.
Line 29: Including “development” about expression studies would help readers know general biological process that expression studies are focusing on.
Line 30: There is an orphaned “ , “ at start of this line. Need to delete a space.
Line 31: If “OsALOG” is referring to genes, then it should be italicized.
Introduction:
Line 45: Delete “s”, so that it reads, “…OsG1 can affect the expression…”
Line 50: Add “space” after “genes” and before “(Li et al….)”.
Methods:
Line 100: Need to either explain more than just “planted in a greenhouse” for how the Indica rice was grown (soil type, time of year, etc…) OR cite a paper that more thoroughly explains how the rice plants were grown.
Line 109: Capital “F” to start sentence, and lower-case “w” for “we” in middle of sentence.
Line 114: Should be “…were downloaded …” (Not “was”).
Lines 131 – 135. All three of these are incomplete sentence. Rewrite to be complete sentences.
Lines 150 – 155. All of the sentences in Section 2.5 are incomplete. Rewrite to be complete sentences for this entire section.
Line 161: Explain what “R” electrophoresis is. Having done agarose gel electrophoresis for 40 years, I’m not sure what this is.
Lines 165-171: How many independent isolated RNA replicates were used for qRT-PCR? Seems possible three, from RNA-seq description above, but this needs to be explicitly stated both here and in Figure 6.
Results:
Line 176: Delete “s” at end of genomes, so that it reads, “…in each genome.”
Lines 174-195 as well as in Figure 1 and Supplemental Figures: Authors need to make it clear which Os genes / proteins are from Indica and which are from Japonica rice species. This would help reader know if genes being discussed are orthologs or paralogs. Not clear to me the best way to do this, but it is important that the authors do so.
Line 193: Regarding Phylogenetic group 3 (Arabidopsis genes only), authors should mention any functional and gene expression data / reports for these. Otherwise, it seems they are mostly being ignored.
Figures 1, 2, 3 and 4: The versions of the figures provided to reviewers were impossible to read much of the text, letters, and numbers. I realize if in an online published versions sometimes the figures can be enlarged. But, as was provided to reviewers and if this is the format/size figures would be shown, much of the text is not legible. This must be fixed before it could be published.
Figure 1: Also, make sure number units are included. As is, it is not clear if this is shown or not, since font (size 2?) is so small and impossible to read.
Figure 1 Legend: add “genes and” to last sentence so that it reads, “…length of genes and proteins can be estimated …”
Line 206: Add “exon” to emphasize this, so that it reads, “… single coding sequence exon.”
Line 208: Need to add a reference/citation after the statement, “compared to other genes in rice and Arabidopsis.”
Line 210: Replace “genes” with “proteins” for Figure 1B and ALOG and IRB domains are for the predicted proteins and not the genes directly.
Line 216: As mentioned above (Line 174-195), make it clear that OsG1 is for Indica rice while OsMH63G1 is Japonica rice.
Line 217: add “regions” to this, so that it is clear there are additional amino acid sequence regions and not just two additional “amino acids”. As it is written, this is not clear.
Line 226: Change “mutation” to “sequence variant”. Since it is not known which is ancestral and which is derived, mutation implies a derived defective mutation/change.
Lines 236-237: Should add “possibly” perform different functions, for their functions are for certain known so it is only speculation that the changes might result in different functions.
Lines 254-261: These are interesting protein characteristics (% non-polar amino acids; high % of Ala,Gly and Pro; net charge, and isoelectric focus point), but they only really interesting if these are different than say “typical” more “most” proteins or comparable to other known transcription factors. Thus, authors should mention what the average rice protein and transcription factor has for these protein characteristics and citations/references for these. This information and citations could be provided either in Results or in the Discussion (for currently, these are not mentioned in Discussion).
Figure 2: As mentioned above, font size is very small and hard to read. Also, replace “Ratio” with “Percentage” for heading at bottom of figure. They are different.
Figure 2 Legend: Needs to emphasize “percentage” for the scale bar on right side of figure. Extra spaces before last “ . “ at end.
Line 266: Delete “Firstly.” Not needed since it is the first sentence.
Line 275: Need to specific Japonica rice in text about synteny data in Figure 3, for it only says “Japonica”. Note, if Indica is included in these analysis that needs to be made clear in the text and figure plus figure legend.
Figure 3: Similar issue to Figures 1 and 2, text is way too small to read, as is.
Figure 3 Legend: Should read, “Red lines indicate homologous gene pairs.”
Lines 288-305 (Section 3.5): Authors need to make is clear that these are just predicted cis-elements, for unless they are directly tested experimentally, it is not clear that they actually function for these predictions. To do this, be sure to include that these are “predicted cis-elements”, etc…
Line 304-305: As mentioned immediately above, in this overall statement at end of paragraph, be sure to make it clear these are “predicted” elements that “might” or “could possibly” lead to distinct expression patterns.
Figure 4: As already mentioned several times. Font sizes are too small and impossible to read. This is the worst of all the figures. PLUS, there is not “A” or “B” labels for the different panels on this figure, yet A and B are mentioned in the Figure Legend.
Figure 4 Legend: need to italicize the “cis” for all uses in the Legend.
Lines 312-316: Need to emphasize that the expression data are for Indica rice only.
Figure 5 Legend: A bit more explanation about the normalization. “Row scale” normalization is perhaps not the most common or best way to say this. Z-score normalization across a row to allow row-to-row comparisons. This could also be included in the Methods.
Figure 6: Best to show the qRT-PCR graphs in same order as those for Figure 5 RNA-seq. It will help readers compare the two. Also, it is a bit concerning that the “Y axes” scales differ for each gene/RNA. It is clear why, so that readers can see small differences, but a reader needs to notice this to know that absolute levels differ. This could be best done by emphasizing that the scales are different for each RNA in the text and Figure Legend.
Figure 6 Legend: As mentioned in Methods, it needs to be made clear how many independent isolated RNA replicates were used for qRT-PCR and that the bars show average of these (three ???). Make this clear.
Supplemental Figures: Be sure to use the same names for genes/proteins in the supplemental figures and tables as those used in the “main paper” and to make clear if genes/proteins are from Indica or Japonica.
Discussion:
Lines 354-355: Need to add a reference/citation of key papers about this, likely just citing already existing references from introduction.
Lines 364-371: As mentioned above (Lines 254-261), these are interesting protein characteristics (% non-polar amino acids; high % of Ala,Gly and Pro; net charge, and isoelectric focus point), but they only really interesting if these are different than say “typical” more “most” proteins or comparable to other known transcription factors. Thus, authors should mention what the average rice protein and transcription factor has for these protein characteristics and citations/references for these.
Lines 388-389 Authors state that OsG1L2 is “3-4 times higher” expression than OsG1L1 and OsG1L5/TAW1, but the Figure 6 qRT-PCR data for OsG1L1 is roughly the same relative values (~11 – 13) as for OsG1L5/TAW1. Either delete or change this statement in the Discussion to match the data or provide further evidence for why it is the case, if it really is 3-4 times higher.
Line 389: Authors state that OsG1L2 is expressed “to a certain extent in each spikelet organ”. But, Figure 6 qRT-PCR data for OsG1L2 only shows small level of expression in the sterile lemma (sl) and none in remaining four organs (pl, le, fl, and br). Either delete or change this statement in the Discussion to match the data or provide further evidence for why it is the case, if it really is 3-4 times higher.
Related to above to items (Lines 388-389): Statements about protein/gene function from absolute RNA levels need to be cautionary for there are plenty of examples in plants (and other organisms) where absolute RNA levels do NOT correlate with proteins levels.
Line 394: Delete “highly” at end of this line, for the OsG1L6 RNA (qRT-PCR) data suggest this RNA is medium to lower absolute expression levels, not “high” levels of expression.
Lines 406-408: Data in Figures 5 and 6 do not support this conclusion. OsG1 has high levels of RNA expression at “S1” young panicle stage, just like OsG1L5 and OsG1L2, so no evidence that argues that OsG1 expression begins to be expressed after these others. Either delete or change this statement in the Discussion to match the data or provide further evidence for why it is the case, if it really is 3-4 times higher.
Lines 408-410: Again, not clear the Figure 5 and 6 evidence supports this statement. OsG1L6 shows clear, but low, expression in palea and lemma. Please either delete statement of provide explanation of data.
References (no Lines):
I was not able to find the Ikeda et al (2004) reference cited anywhere within the manuscript text, figures, or in the supplementary data. Either delete from References from the list or be sure to add the “in text” citation to correct location.
Comments on the Quality of English LanguageNo additional comments, for I have made all of my comments about English language above.
Author Response
Reviewer 2: Comments and Suggestions for Authors
The authors have bioinformatic of gene and predicted proteins along with RNA expression (RNA-seq and qRT-PCR) data for the family of ALOG (Arabidopsis LSH1 and Oryza G1) genes and proteins in rice. They cite previous reports from a variety of groups that suggest the function of these genes/proteins are transcriptional regulators of development in rice. The authors themselves do not provide additional “functional” data, but instead provide bioinformatic phylogenetic, gene structure, predicted promoter regulatory cis-elements, general protein structure, and amino acid bias data that might relate to function and functional differences across the 10 genes in rice. Further, they provided RNA expression data (both RNA-seq and confirmatory qRT-PCR data) for the developmental expression (at the RNA level) for the same 10 genes in Indica rice (did not test Japonica rice) and in conclusion related expression patterns to phylogenetic, protein structure/aa sequence groups and compared to predicted function from cited reports.
Understanding rice development is critical for basic plant biology reasons and for the world-wide importance of rice for human nutrition, thus there is value in understanding these genes and the encoded proteins. In general, the methods and data analysis of the authors were reasonable and consistent with standard practices, but there were no profoundly new discoveries or conclusions as to function and roles of ALOG genes/proteins. However, the provided gene/protein characterizations and expression data do add some to the understanding of ALOG’s in rice. I take issue with some minor data interpretation of the expression data by the authors in the Discussion compared to what the provided data show, as mentioned below. On a positive note, it was good to see that the RNA-seq data closely matched the qRT-PCR data and that the authors did the important confirmation with the qRT-PCR analysis.
The biggest general scientific concern I have is that there are functional data (from previously cited reports) only for the Phylogenetic Group 1 genes/proteins (see Figure 1) that show them to be transcriptional factors, and no functional data (here or in other reports) for Phylogenetic Group 2 (rice and Arabidopsis genes) or Group 3 (Arabidopsis genes only). The authors seem to be making the general assumption that Group 2 and 3 proteins would also be transcription factors, but again no data to show that. Any additional functional data/evidence information about Phylogenetic Groups 2 and 3 genes/proteins would help this paper significantly.
Response:Thank you for this constructive suggestion. We appreciate the opportunity to clarify our rationale for focusing on phylogenetic analysis in this section.
As noted in the Introduction (second paragragh), six Arabidopsis ALOG genes (AtLSH1/2/3/4/8/10) have been functionally characterized. These genes are known to regulate meristem development and organ boundary formation. The main objective of this section was to analyze: a) The clustering patterns of different rice ALOG genes. b) The evolutionary divergence between rice and Arabidopsis ALOG homologs. The functional aspects of these genes have been discussed in both the Introduction and Discussion sections.
A second general issue that comes up throughout manuscript text, figures, and supplemental data is that authors should make it clear what Os genes / proteins are from Indica and which are from Japonica. This would help reader know if genes being discussed are orthologs or paralogs.
Response:Thank you for your suggestion. We have included the donor species information of the downloaded genomic data in “Materials and Methods section 2.2” and added corresponding annotations in Table S1 for clarification. . (Page 3, Lines 119)
In addition to this concern, there a number of writing issues/errors, missing methods information, figure layout problems, and some gene expression data interpretation that I feel the authors need to resolve before this could be published. See my specific comments below for these issues.
Response:We sincerely appreciate your thorough review and valuable suggestions. We have carefully addressed all raised concerns, including: Writing Issues, Methodological Gaps, Figure Layout and Data Interpretation.
Specific point-by-point responses follow below. We are grateful for your time and expertise, which have significantly strengthened our manuscript.
Lines 15 - 17: It would help readers if some the predicted function(s) for transcription factors for ALOG proteins is mentioned in Abstract.
Response:We appreciate your suggestion. We have incorporated the statement "The ALOG family primarily functions as transcription factors" in the abstract section (Page 1, Lines 15-18), with the newly added text highlighted for emphasis.
Line 20: Here “ALOG” is refereeing to genes, so it should be italicized.
Response:The text has been revised as italicized.
Lines 26: Mention that these are predicted “promoter” elements in the genes, versus elements that might control RNA process or some other genetic process.
Response:The text has been revised as suggestion and highlighted (Page 1, Lines 27).
Line 29: Including “development” about expression studies would help readers know general biological process that expression studies are focusing on.
Response:We have incorporated the statement " The expression patterns associated with rice floral development of OsALOG genes " in the abstract section (Page 1, Lines 29-30), with the newly added text highlighted for emphasis.
Line 30: There is an orphaned “ , “ at start of this line. Need to delete a space.
Response:The text has been revised as suggestion.
Line 31: If “OsALOG” is referring to genes, then it should be italicized.
Response:The text has been revised as suggestion. (Page 2, Lines 33)
Introduction:
Line 45: Delete “s”, so that it reads, “…OsG1 can affect the expression…”
Response:The text has been revised as suggestion. (Page 2, Lines 47)
Line 50: Add “space” after “genes” and before “(Li et al….)”.
Response:The textual mistakes has been revised as suggestion. (Page 2, Lines 52)
Methods:
Line 100: Need to either explain more than just “planted in a greenhouse” for how the Indica rice was grown (soil type, time of year, etc…) OR cite a paper that more thoroughly explains how the rice plants were grown.
Response:The description of cultivation conditions has been added as suggested. (Page 3, Lines 102-108)
Line 109: Capital “F” to start sentence, and lower-case “w” for “we” in middle of sentence.
Response:The textual mistakes has been revised as suggestion. (Page 3, Lines 117)
Line 114: Should be “…were downloaded …” (Not “was”).
Response:The Grammatical errors has been revised as suggestion. (Page 3, Lines 122)
Lines 131 – 135. All three of these are incomplete sentence. Rewrite to be complete sentences.
Response:The Grammatical errors has been revised as suggestion. (Page 4, Lines 139-146)
Lines 150 – 155. All of the sentences in Section 2.5 are incomplete. Rewrite to be complete sentences for this entire section.
Response:The Grammatical errors has been revised as suggestion. (Page 4, Lines 161-169)
Line 161: Explain what “R” electrophoresis is. Having done agarose gel electrophoresis for 40 years, I’m not sure what this is.
Response:Thank you for your careful reading of our manuscript. We sincerely apologize for any confusion caused by the wording in Line 161.
We have revised the manuscript to explicitly state "RNA electrophoresis "instead of just "R electrophoresis " to prevent any future misunderstanding. (Page 4, Lines 175)
Lines 165-171: How many independent isolated RNA replicates were used for qRT-PCR? Seems possible three, from RNA-seq description above, but this needs to be explicitly stated both here and in Figure 6.
Response:Thank you for your careful review and valuable suggestion. Both the RNA-seq analysis and qRT-PCR validation experiments were performed using three independent biological replicates for each sample. As suggested, we have now explicitly stated this information in page 4 line 171-172 and Figure 6.
Results:
Line 176: Delete “s” at end of genomes, so that it reads, “…in each genome.”
Response:Thank you for your careful review, the Grammatical errors has been revised as suggestion. (Page 5, Lines 191)
Lines 174-195 as well as in Figure 1 and Supplemental Figures: Authors need to make it clear which Os genes / proteins are from Indica and which are from Japonica rice species. This would help reader know if genes being discussed are orthologs or paralogs. Not clear to me the best way to do this, but it is important that the authors do so.
Response:Thank you for your suggestion. We have included the donor species information of the downloaded genomic data in “Materials and Methods section 2.2” and added corresponding annotations in Table S1 for clarification. . (Page 3, Lines 119)
Line 193: Regarding Phylogenetic group 3 (Arabidopsis genes only), authors should mention any functional and gene expression data / reports for these. Otherwise, it seems they are mostly being ignored.
Response:Thank you for this constructive suggestion. We appreciate the opportunity to clarify our rationale for focusing on phylogenetic analysis in this section.
As noted in the Introduction (second paragragh), six Arabidopsis ALOG genes (AtLSH1/2/3/4/8/10) have been functionally characterized. These genes are known to regulate meristem development and organ boundary formation. The main objective of this section was to analyze: a) The clustering patterns of different rice ALOG genes. b) The evolutionary divergence between rice and Arabidopsis ALOG homologs. The functional aspects of these genes have been discussed in both the Introduction and Discussion sections.
Figures 1, 2, 3 and 4: The versions of the figures provided to reviewers were impossible to read much of the text, letters, and numbers. I realize if in an online published versions sometimes the figures can be enlarged. But, as was provided to reviewers and if this is the format/size figures would be shown, much of the text is not legible. This must be fixed before it could be published.
Figure 1: Also, make sure number units are included. As is, it is not clear if this is shown or not, since font (size 2?) is so small and impossible to read.
Response:Thank you for your suggestion. We have provided the original PDF files of both the main and supplementary figures to the editorial office, to facilitate clarity assessment and layout adjustments during the editing process.
Figure 1 Legend: add “genes and” to last sentence so that it reads, “…length of genes and proteins can be estimated …”
Response:Thank you for your careful review, the sentence has been revised as suggestion. (Page 5, Lines 218)
Line 206: Add “exon” to emphasize this, so that it reads, “… single coding sequence exon.”
Response:Thank you for your careful review, the “exon” has added as suggestion. (Page 6, Lines 224)
Line 208: Need to add a reference/citation after the statement, “compared to other genes in rice and Arabidopsis.”
Response:Thank you for your careful review, the sentence has been revised as suggestion. (Page 6, Lines 226-227)
Line 210: Replace “genes” with “proteins” for Figure 1B and ALOG and IRB domains are for the predicted proteins and not the genes directly.
Response:Thank you for your careful review, the “genes” has replaced with “proteins” as suggestion. (Page 6, Lines 229)
Line 216: As mentioned above (Line 174-195), make it clear that OsG1 is for Indica rice while OsMH63G1 is Japonica rice.
Response:Thank you for your suggestion. The prefix "Os" represents Oryza sativa ssp. japonica (cultivar Nipponbare), the prefix "OsMH63" represents Oryza sativa ssp. indica (cultivar Minghui 63), and the prefix "At" denotes Arabidopsis thaliana (thale cress), the same below.
Line 217: add “regions” to this, so that it is clear there are additional amino acid sequence regions and not just two additional “amino acids”. As it is written, this is not clear.
Response:Thank you for your careful review, the errors has been revised as suggestion. (Page 6, Lines 235)
Line 226: Change “mutation” to “sequence variant”. Since it is not known which is ancestral and which is derived, mutation implies a derived defective mutation/change.
Response:Thank you for your careful review, the “mutation” has replaced with “sequence variant” as suggestion. (Page 6, Lines 254)
Lines 236-237: Should add “possibly” perform different functions, for their functions are for certain known so it is only speculation that the changes might result in different functions.
Response:Thank you for your careful review, the “possibly” has added as suggestion. (Page 6, Lines 255)
Lines 254-261: These are interesting protein characteristics (% non-polar amino acids; high % of Ala,Gly and Pro; net charge, and isoelectric focus point), but they only really interesting if these are different than say “typical” more “most” proteins or comparable to other known transcription factors. Thus, authors should mention what the average rice protein and transcription factor has for these protein characteristics and citations/references for these. This information and citations could be provided either in Results or in the Discussion (for currently, these are not mentioned in Discussion).
Response:We sincerely appreciate your valuable suggestion. Currently, research on the functions of IDRs (intrinsically disordered regions) in plants remains at an exploratory stage. In our study, we found that the IDRs of rice ALOG proteins are enriched with three nonpolar amino acid residues and one polar amino acid residue, which shows some inconsistency with previous findings. Therefore, the specific functions and characteristics of IDRs in rice ALOG proteins require further in-depth investigation. In this paper, we systematically analyzed three key physicochemical properties of rice IDPs: the proportion of nonpolar amino acids, the net charge (reflecting acid-base properties), and the isoelectric point. These analyses were conducted to establish a fundamental database for future research in this field. We greatly appreciate your insightful comment, which has helped us better articulate our work. Please let us know if any additional clarification would be helpful.
Figure 2: As mentioned above, font size is very small and hard to read. Also, replace “Ratio” with “Percentage” for heading at bottom of figure. They are different.
Response:Thank you for your careful review, the “Ratio” in Figure 2 has replaced with “Percentage” as suggestion.
Figure 2 Legend: Needs to emphasize “percentage” for the scale bar on right side of figure. Extra spaces before last “ . “ at end.
Response:Thank you for your careful review, the errors has been revised as suggestion.
Line 266: Delete “Firstly.” Not needed since it is the first sentence.
Response:Thank you for your careful review, the errors has been revised as suggestion.
Line 275: Need to specific Japonica rice in text about synteny data in Figure 3, for it only says “Japonica”. Note, if Indica is included in these analysis that needs to be made clear in the text and figure plus figure legend.
Response:Thank you for your suggestion. the “Oryza sativa Japonica” has added as note . (Page 8, Lines 304, 305)
Figure 3: Similar issue to Figures 1 and 2, text is way too small to read, as is.
Response:Thank you for your suggestion. We have provided the original PDF files of both the main and supplementary figures to the editorial office, to facilitate clarity assessment and layout adjustments during the editing process.
Figure 3 Legend: Should read, “Red lines indicate homologous gene pairs.”
Response:Thank you for your careful review, the errors has been revised as suggestion.
Lines 288-305 (Section 3.5): Authors need to make is clear that these are just predicted cis-elements, for unless they are directly tested experimentally, it is not clear that they actually function for these predictions. To do this, be sure to include that these are “predicted cis-elements”, etc…
Response:Thank you for your suggestion. We have revised the first sentence of this section to emphasize the predictive nature of the cis-acting elements, thereby maintaining academic rigor. (Page 8, Lines 307-309)
Line 304-305: As mentioned immediately above, in this overall statement at end of paragraph, be sure to make it clear these are “predicted” elements that “might” or “could possibly” lead to distinct expression patterns.
Response:Thank you for your suggestion. We have revised the first sentence of this section to emphasize the predictive nature of the cis-acting elements, thereby maintaining academic rigor. (Page 8, Lines 307-309)
Figure 4: As already mentioned several times. Font sizes are too small and impossible to read. This is the worst of all the figures. PLUS, there is not “A” or “B” labels for the different panels on this figure, yet A and B are mentioned in the Figure Legend.
Response:We sincerely apologize for the issues with Figure 4 and appreciate you bringing these important matters to our attention. we have now submitted the original PDF files of all figures to the editorial office, which will allow for proper font size adjustments and layout modifications during the production process. We have carefully revised Figure 4 by: Adding clear "A" and "B" labels to each corresponding panel. Ensuring these labels match the descriptions in the figure legend.Verifying the visibility of all labels in the resubmitted version.
Figure 4 Legend: need to italicize the “cis” for all uses in the Legend.
Response:We have revised the text errors in Figure 4.
Lines 312-316: Need to emphasize that the expression data are for Indica rice only.
Response:We sincerely appreciate your suggestion. In response, we have explicitly stated in the first paragraph of Results section 3.6 that the RNA-seq samples were derived from the indica rice cultivar Huanghuazhan, to provide full transparency about the biological materials used in our transcriptome analysis.
Figure 5 Legend: A bit more explanation about the normalization. “Row scale” normalization is perhaps not the most common or best way to say this. Z-score normalization across a row to allow row-to-row comparisons. This could also be included in the Methods.
Response:Thank you for your suggestion. In the "Heatmap" function of TBtools, when selecting "row-wise normalization" for clustering, the default standardization method used is Z-score normalization (standard deviation standardization). We have now added a note regarding this in the text description of Figure 5. (page 9 line 372)
Figure 6: Best to show the qRT-PCR graphs in same order as those for Figure 5 RNA-seq. It will help readers compare the two. Also, it is a bit concerning that the “Y axes” scales differ for each gene/RNA. It is clear why, so that readers can see small differences, but a reader needs to notice this to know that absolute levels differ. This could be best done by emphasizing that the scales are different for each RNA in the text and Figure Legend.
Response:Thank you for your suggestion. We have rearranged the qRT-PCR bar chart according to the clustering order in Figure 5, with the fill colors modified to match those of Group â… and Group â…¡ in Figure 5, and added corresponding annotations in Figure 6. (page 10 line 385-386)
Figure 6 Legend: As mentioned in Methods, it needs to be made clear how many independent isolated RNA replicates were used for qRT-PCR and that the bars show average of these (three ???). Make this clear.
Supplemental Figures: Be sure to use the same names for genes/proteins in the supplemental figures and tables as those used in the “main paper” and to make clear if genes/proteins are from Indica or Japonica.
Response:Thank you for your careful review. Both the RNA-seq analysis and qRT-PCR validation experiments were performed using three independent biological replicates for each sample. As suggested, we have now explicitly stated this information in page 4 line 171-172 and Figure 6.
To ensure clarity and consistency, we adopted the official gene names from UniProt for labeling ALOG genes in both the expression profiles and qPCR bar charts. This avoids potential confusion arising from alternative nomenclature. Our multiple sequence alignment revealed minimal amino acid sequence divergence between indica and japonica ALOG proteins, particularly within the conserved ALOG domains.
Additionally, we explicitly stated that all experimental samples were derived from the indica cultivar 'Huanghuazhan'. For standardization, we also corrected the gene names in Figure S6 to match those used in other supplementary figures.
Discussion:
Lines 354-355: Need to add a reference/citation of key papers about this, likely just citing already existing references from introduction.
Response:Thank you for your careful review. We have added a reference/citation of key papers about ALOG family proteins are generally considered to be transcription factors.
Lines 364-371: As mentioned above (Lines 254-261), these are interesting protein characteristics (% non-polar amino acids; high % of Ala,Gly and Pro; net charge, and isoelectric focus point), but they only really interesting if these are different than say “typical” more “most” proteins or comparable to other known transcription factors. Thus, authors should mention what the average rice protein and transcription factor has for these protein characteristics and citations/references for these.
Response:We sincerely appreciate your valuable suggestion. Currently, research on the functions of IDRs (intrinsically disordered regions) in plants remains at an exploratory stage. In our study, we found that the IDRs of rice ALOG proteins are enriched with three nonpolar amino acid residues and one polar amino acid residue, which shows some inconsistency with previous findings. Therefore, the specific functions and characteristics of IDRs in rice ALOG proteins require further in-depth investigation. In this paper, we systematically analyzed three key physicochemical properties of rice IDPs: the proportion of nonpolar amino acids, the net charge (reflecting acid-base properties), and the isoelectric point. These analyses were conducted to establish a fundamental database for future research in this field. We greatly appreciate your insightful comment, which has helped us better articulate our work. Please let us know if any additional clarification would be helpful.
Lines 388-389 Authors state that OsG1L2 is “3-4 times higher” expression than OsG1L1 and OsG1L5/TAW1, but the Figure 6 qRT-PCR data for OsG1L1 is roughly the same relative values (~11 – 13) as for OsG1L5/TAW1. Either delete or change this statement in the Discussion to match the data or provide further evidence for why it is the case, if it really is 3-4 times higher.
Response:We sincerely appreciate your valuable suggestion. We recognize that claiming '3-4 times higher' expression was inaccurate, and have consequently rephrased the expression profile of OsG1L2 with more precise language. (Page 11, Lines 425-428)
Line 389: Authors state that OsG1L2 is expressed “to a certain extent in each spikelet organ”. But, Figure 6 qRT-PCR data for OsG1L2 only shows small level of expression in the sterile lemma (sl) and none in remaining four organs (pl, le, fl, and br). Either delete or change this statement in the Discussion to match the data or provide further evidence for why it is the case, if it really is 3-4 times higher.
Response:We sincerely appreciate your valuable suggestion. We recognize that claiming 'to a certain extent in each spikelet organ' expression was inaccurate, and have consequently rephrased the expression profile of OsG1L2 with more precise language. (Page 11, Lines 425-428)
Related to above to items (Lines 388-389): Statements about protein/gene function from absolute RNA levels need to be cautionary for there are plenty of examples in plants (and other organisms) where absolute RNA levels do NOT correlate with proteins levels.
Response:We fully agree with this important point. We have removed any claims about protein function based solely on RNA levels and added a cautionary note about potential transcript-protein discrepancies.
Line 394: Delete “highly” at end of this line, for the OsG1L6 RNA (qRT-PCR) data suggest this RNA is medium to lower absolute expression levels, not “high” levels of expression.
Response:Thank you for your careful review, the “highly” have been deleted.
Lines 406-408: Data in Figures 5 and 6 do not support this conclusion. OsG1 has high levels of RNA expression at “S1” young panicle stage, just like OsG1L5 and OsG1L2, so no evidence that argues that OsG1 expression begins to be expressed after these others. Either delete or change this statement in the Discussion to match the data or provide further evidence for why it is the case, if it really is 3-4 times higher.
Lines 408-410: Again, not clear the Figure 5 and 6 evidence supports this statement. OsG1L6 shows clear, but low, expression in palea and lemma. Please either delete statement of provide explanation of data.
Response:We sincerely appreciate your suggestion regarding the temporal expression patterns of G1 and OsG1L1/2/5/6. We would like to clarify that our proposed sequence of expression timing was based on logical speculation rather than direct experimental evidence (such as in situ hybridization results). Therefore, the original use of "summarize" was indeed inappropriate, and we have revised the wording accordingly.
Regarding the hypothesis that G1 expression occurs after OsG1L1/2/5, this inference was drawn from previous studies (Liu et al., 2016; Yoshida et al., 2013; Beretta et al., 2013). Furthermore, based on the gradually increasing expression pattern of OsG1L6 observed in our study, we speculate that its expression initiation might occur even later than G1. We acknowledge that our original wording did not sufficiently emphasize the speculative nature of these conclusions, and we have now carefully modified the relevant text to better reflect this.
We greatly value your insightful comments, which have helped improve the accuracy and clarity of our manuscript. Please let us know if any additional refinements would be helpful.
References (no Lines):
I was not able to find the Ikeda et al (2004) reference cited anywhere within the manuscript text, figures, or in the supplementary data. Either delete from References from the list or be sure to add the “in text” citation to correct location.
Response:Thank you for your careful review, We confirm that the reference (Ikeda et al., 2004) was cited as the standard for determining sampling timepoints in our transcriptome sequencing experiments. We have now reformatted all references throughout the manuscript according to the journal's citation style guidelines.
